# THE COMPLEXITY OF TWO-TEAM POLYMATRIX GAMES WITH INDEPENDENT ADVERSARIES

**Alexandros Hollender**
University of Oxford
alexandros.hollender@cs.ox.ac.uk

**Gilbert Maystre**
EPFL
gilbert.maystre@epfl.ch

**Sai Ganesh Nagarajan**
Zuse Institute Berlin
nagarajan@zib.de

## ABSTRACT

Adversarial multiplayer games are an important object of study in multiagent learning. In particular, polymatrix zero-sum games are a multiplayer setting where Nash equilibria are known to be efficiently computable. Towards understanding the limits of tractability in polymatrix games, we study the computation of Nash equilibria in such games where each pair of players plays either a zero-sum or a coordination game. We are particularly interested in the setting where players can be grouped into a small number of teams of identical interest. While the three-team version of the problem is known to be PPAD-complete, the complexity for two teams has remained open. Our main contribution is to prove that the two-team version remains hard, namely it is CLS-hard. Furthermore, we show that this lower bound is tight for the setting where one of the teams consists of multiple independent adversaries. On the way to obtaining our main result, we prove hardness of finding any stationary point in the simplest type of non-convex-concave min-max constrained optimization problem, namely for a class of bilinear polynomial objective functions.

## 1 INTRODUCTION

Game theory is a fundamental tool to encode strategic agent interactions and has found many applications in the modern AI landscape such as Generative Adversarial Networks (Goodfellow et al., 2020), obtaining agents with expert level play in multiplayer games such as Starcraft and Quake III (Vinyals et al., 2019; Jaderberg et al., 2019) and superhuman performance in poker (Brown & Sandholm, 2018; 2019). Computing a Nash equilibrium or a saddle point (when considering general minmax optimization problems) is a computational task of central importance in these applications. The celebrated minmax theorem of von Neumann & Morgenstern (1947) established that two-player zero-sum games have efficient algorithms. However, it was shown that three-player zero-sum games (Daskalakis et al., 2009) or two-player general games (Chen et al., 2009) are computationally intractable (formally PPAD-hard) and the hardness is also known to hold for computing approximations.

Consequently, Daskalakis & Papadimitriou (2009) proposed a tractable class of multiplayer zero-sum games, where the players are placed on the nodes of a graph and play a matrix zero-sum game with each adjacent player. In this setting, the total utility that a player gets is the sum of the utilities from each game that they participate in. It is to be highlighted that removing the zero-sum game assumption between each player makes the problem hard. Indeed, computing Nash equilibria in general polymatrix games is known to be PPAD-hard (Daskalakis et al., 2009; Chen et al., 2009; Rubinstein, 2018; Deligkas et al., 2024). Cai & Daskalakis (2011) studied an intermediate setting where every edge of the polymatrix game can be either a zero-sum game or a coordination game. They showed PPAD-hardness, even for the special case where the players can be grouped into **three**

---

Authors contributed equally to this work.

teams, such that players within the same team play coordination games, and players in different teams play zero-sum games.

**Adversarial Two-team Games With a Single Adversary:**    The general notion of adversarial *team* games introduced by  von Stengel & Koller (1997) studies two teams that are playing a zero-sum game with each other, meaning each team member gets the *same* payoff as the whole team and the sum of the team payoffs is zero. The primary motivation here is to study strategies for companies against adversaries. What makes the "team" aspect special is that the team members cannot coordinate their actions and must play independent mixed strategies. This captures imperfect coordination within companies. Indeed, if the teams instead had perfect coordination, then the setting would simply reduce to a two player zero-sum game. Von Stengel and Koller showed that there exists a team maxmin equilibrium, that can be extended to a Nash equilibrium for this setting when the team is playing against a *single* adversary. Moreover, they showed that this is the *best* Nash equilibrium for the team, thereby alleviating equilibrium selection issues. However, it was later shown that finding a team maxmin equilibrium is in fact FNP-hard and the problem does not become easier if one allows approximate solutions (Hansen et al., 2008; Borgs et al., 2008). Recently,  Anagnostides et al. (2023) studied the *single* adversary setting, and were able to show that finding a Nash equilibrium in this setting is in fact CLS-complete. The CLS-hardness immediately follows from the work of Babichenko & Rubinstein (2021), but importantly it requires a sufficiently general game structure and so does not apply to the polymatrix setting. The CLS-membership on the other hand applies to any adversarial game with a single adversary. The main idea is to obtain a Nash equilibrium from an approximate stationary point of the max Moreau envelope of the function $x \mapsto \max_{y \in \mathcal{Y}} U(x, y)$ (where $x$ is the min variable, $y$ is the max variable and $U$ is the payoff function).

**Connections to Complexity of Minmax Optimization:**    Two-team games are a special case of general nonconvex-nonconcave constrained minmax problems. Daskalakis et al. (2021) recently studied this general setting and showed that finding a stationary point is PPAD-complete. Crucially, their PPAD-hardness only applies when the constraint sets are *coupled* between the min and the max player. However, games usually induce minmax problems with *uncoupled* constraints. The complexity of the problem for uncoupled constraints remains open, although it is known to be CLS-hard, since it is at least as hard as finding stationary points of standard non-convex minimization problems (Fearnley et al., 2022). As we discuss below, our results also have implications for uncoupled minmax optimization, where we obtain a CLS-hardness result for a particularly simple family of objective functions. We note that  Li et al. (2021) showed a *query* lower bound of $\Omega(\frac{1}{\varepsilon^2})$ for smooth nonconvex-strongly-concave minmax optimization problems, but these results do not apply to the simple objective functions that we study (and which can only be studied from the perspective of computational complexity).

**Connections to Multiagent Learning:**    From a learning dynamics perspective, qualitative results focus on understanding the limit behavior of certain no-regret learning dynamics in polymatrix games. In particular, some works focus on obtaining asymptotic convergence guarantees for Q-learning and its variants (Leonardos et al., 2021; Hussain et al., 2023). In a similar vein, some other works studied the limit behaviors of replicator dynamics for polymatrix games, particularly with zero-sum and coordination edges (Nagarajan et al., 2018; 2020). In these works, the focus was on trying to identify network topologies under which the learning dynamics exhibited simple (non-chaotic) behaviors. Surprisingly, there were works that could obtain non-asymptotic convergence guarantees using discrete time algorithms in multiagent reinforcement learning as well, with  Leonardos et al. (2022) establishing convergence to Nash policies in Markov potential games. In adversarial settings, Daskalakis et al. (2020) studied independent policy gradient and proved convergence to Nash policies. Moreover, some recent works establish convergence to Nash policies in Markov zero-sum team games (Kalogiannis et al., 2023) and Markov polymatrix zero-sum games (Kalogiannis & Panageas, 2023). This further establishes the need to theoretically study the computational challenges in the simplest polymatrix settings which allow for *both* zero-sum and coordination edges, in order to understand convergence guarantees in more complicated multiagent reinforcement learning scenarios.

This leads us to the following main question that had been open from the work of  Cai & Daskalakis (2011).

*What is the complexity of finding Nash equilibria in two-team zero-sum polymatrix games?*

## 1.1 OUR CONTRIBUTIONS

Our main contribution is the following computational hardness result.

**Theorem 1.1** (Informal). *It is* CLS-*hard to find an approximate Nash equilibrium of a two-team zero-sum polymatrix game, even when one of the teams does not have any internal edges.*

Our result is incomparable to the CLS-hardness result proved by Anagnostides et al. (2023) (which essentially immediately follows from the work of Babichenko & Rubinstein (2021)). On the one hand, our result is stronger because it applies to games with a simpler structure, namely polymatrix games, whereas their result only applies to the more general class of *degree-5 polytensor* games. On the other hand, our (hardness) result is weaker because it requires the presence of multiple adversaries, instead of just a single adversary. The case of two-team polymatrix games with a single adversary remains open. If one could prove CLS-hardness for that version of the problem, then this would constitute a strengthening of both our result and the result of Anagnostides et al. Proving Theorem 1.1 requires a novel construction that takes advantage of the max-variables to introduce additional constraints on the min-variables. This step is crucial to obtain the required bilinear form for the objective and constitutes our main technical contribution.

As our second contribution, we complement the hardness result in Theorem 1.1 by showing that the problem is in fact CLS-complete in this particular case where the adversaries are independent (i.e., when there are no internal edges in the second team). Namely, the problem of finding an approximate Nash equilibrium in a two-team zero-sum polymatrix game with multiple independent adversaries lies in the class CLS. The polymatrix setting allows us to provide a simple proof of this fact, in particular avoiding the use of more advanced machinery, such as the Moreau envelope used in the CLS-membership of Anagnostides et al. (2023). We note that if the adversaries are not independent, then the problem is only known to lie in PPAD, and it remains an important open problem to determine whether it also lies in CLS or is in fact PPAD-complete.

Going back to our main result, Theorem 1.1, we note that it also has some interesting consequences for minmax optimization in general. Namely, we obtain that computing a stationary point, i.e., a Karush-Kuhn-Tucker (KKT) point of

$$\min_{x \in [0,1]^n} \max_{y \in [0,1]^m} f(x,y)$$

is CLS-hard, and thus intractable in the worst case, even when $f$ is a bilinear polynomial[1] that is concave in $y$. This is somewhat surprising, as these objective functions are the simplest case beyond the well-known tractable setting of convex-concave.

**The meaning of** CLS**-hardness.** The complexity class CLS was introduced by Daskalakis & Papadimitriou (2011) to capture the complexity of problems that are guaranteed to have a solution both by a fixed point argument and a local search argument. This class is a subset of two well-known classes: PPAD and PLS. While PPAD is mainly known for capturing the complexity of computing Nash equilibria in general games (Daskalakis et al., 2009; Chen et al., 2009), PLS captures the complexity of various hard local search problems, such as finding a locally maximal cut in a graph (Schäffer & Yannakakis, 1991). Recently, following the result by Fearnley et al. (2022) that CLS = PPAD ∩ PLS, it has been shown that the class captures the complexity of computing mixed Nash equilibria in congestion games (Babichenko & Rubinstein, 2021), and KKT points of quadratic polynomials (Fearnley et al., 2024). See Figure 1 for an illustration of the relationship between the classes.

A CLS-hardness result indicates that the problem is very unlikely to admit a polynomial-time algorithm. To be more precise, our results indicate that we should not expect an algorithm to exist which can find an $\varepsilon$-approximate Nash equilibrium in these two-team polymatrix games in time polynomial in $\log(1/\varepsilon)$. In contrast, if the team had *perfect* coordination[2], then our setting reduces to a "star-network" zero-sum game which was shown to have a polynomial time algorithm to compute a Nash equilibrium Daskalakis & Papadimitriou (2009).

---

[1]Previous CLS-hardness results required more general objective functions, namely $f$ had to be a quadratic (non-bilinear) polynomial (Fearnley et al., 2024).

[2]With perfect coordination, one can see that the team effectively acts as a single player.

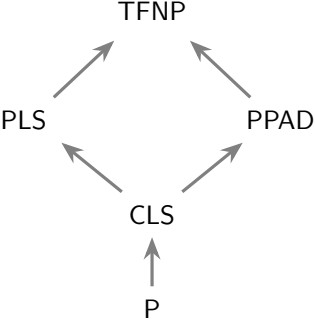

Figure 1: Classes of total search problems. Arrows are used to denote containment. For example, CLS is contained in PLS and in PPAD. The class TFNP contains all total search problems, i.e., problems which are guaranteed to have efficiently checkable solutions. P contains all such problems solvable in polynomial time.

Furthermore, our hardness result for the minmax problem with bilinear polynomial objective functions establishes the fact that we should not expect an algorithm that finds an $\varepsilon$-KKT point in time polynomial in $\log(1/\varepsilon)$. The evidence that CLS is a hard class is supported by various cryptographic lower bounds, which apply to both PPAD and CLS (Bitansky et al., 2015; Choudhuri et al., 2019; Jawale et al., 2021).

## 1.2 Other Related Work

It is worth mentioning that the computation of team maxmin equilibria for the two-team adversarial game has been studied where the team members are allowed to coordinate *ex ante*, which is different from what a polymatrix coordination game would induce, as we study in this paper. There it was shown that the game can be reduced to a two-player game with imperfect-recall (Farina et al., 2018; Celli & Gatti, 2018; Zhang et al., 2022) and that efficient algorithms exist under some assumptions about the players' information sets (Zhang et al., 2021). Finally, similar notions have been studied for extensive-form games too (Zhang & Sandholm, 2022). Please see Appendix B, for additional context with respect to minmax optimization and polymatrix games.

## 2 Preliminaries

### 2.1 Polymatrix games

A polymatrix game (Janovskaja, 1968) is a type of multiplayer game in which the payoff function can be succinctly represented. More precisely, there is a set of players $\mathcal{N}$ and a set of pure strategies $S_i$ for each player $i \in \mathcal{N}$. Moreover, players are represented by the vertices of an undirected graph $G = (\mathcal{N}, E)$ with the intent that a matrix game is played between each pair of players $\{i, j\} \in E$; the pair of payoff matrices being denoted by $A^{i,j} \in \mathbb{R}^{S_i \times S_j}$ and $A^{j,i} \in \mathbb{R}^{S_j \times S_i}$. Players are allowed to randomize over their pure strategies and play a mixed strategy in the probability simplex of their action, which we denote by $\Delta(S_i)$ for player $i$. Hence, a mixed strategy profile is some $x = (x_1, x_2, \ldots, x_{|\mathcal{N}|}) \in \mathcal{X} \coloneqq \times_{i \in \mathcal{N}} \Delta(S_i)$. We also use the standard notation $x_{-i}$ to represent the mixed strategies of all players other than $i$. In polymatrix games, the utility $U_i : \mathcal{X} \to \mathbb{R}$ for player $i \in \mathcal{N}$ is the sum of her payoffs, so that

$$U_i(x) = \sum_{j:\{i,j\} \in E} x_i A^{i,j} x_j$$

where $x_i$ is understood to be $x_i^\top$: we drop the transpose when it is clear from the context for ease of presentation. As a well-defined class of games, polymatrix games always admit a Nash equilibrium (Nash, 1951). In this work, we are interested in *approximate* Nash equilibria which we define next.

**Definition 1.** *Let $\varepsilon \geq 0$ be an approximation guarantee. The mixed strategy profile $\widetilde{x}$ is an $\varepsilon$-approximate Nash equilibrium of the polymatrix game defined above if for any $i \in \mathcal{N}$,*

$$U_i(x_i, \widetilde{x}_{-i}) \leq U_i(\widetilde{x}_i, \widetilde{x}_{-i}) + \varepsilon \quad \forall x_i \in \Delta(S_i)$$

In this paper, we focus on polymatrix games with a particular structure where players are grouped into two competing teams with players within teams sharing mutual interests.

**Definition 2** (Two-team Polymatrix Zero-Sum Game). *A two-team polymatrix zero-sum game is a polymatrix game $\{A^{i,j}\}_{i,j \in \mathcal{N}}$ where the players can be split into two teams $X \cup Y = \mathcal{N}$ so that any game between the two teams is zero-sum and any game within a team is a coordination game. More precisely for any $i, i' \in X$ and $j, j' \in Y$:*

$$A^{i,i'} = \left(A^{i',i}\right)^\top \quad A^{j,j'} = \left(A^{j',j}\right)^\top \quad A^{i,j} = -\left(A^{j,i}\right)^\top$$

*If there is no coordination within team $Y$, that is $A^{j,j'} = 0^{S_j \times S_{j'}}$ for every $j, j' \in Y$, we further say that it is a two-team zero-sum game with independent adversaries.*

In the restricted context of two-teams games, another useful equilibrium concept is that of *team-maxmin equilibria* (von Stengel & Koller, 1997). While originally defined for a single adversary only, it is generalizable to multiple independent adversaries in such a way that any team-maxmin equilibria can be converted to a Nash-equilibrium one efficiently. Unfortunately, similarly to the single-adversary case, such equilibriums suffer from intractability issues and are FNP-hard to compute (Basilico et al., 2017; Hansen et al., 2008; Borgs et al., 2008).

## 2.2 KKT Points of Constrained Optimization Problems

We now turn our attention to solution concepts for optimization problems and in particular of Karush-Kuhn-Tucker (KKT) points. We only define the required notions for the special case where each variable is constrained to be in $[0, 1]$, since this will be sufficient for us. Under those *box constraints* the expression of the Lagrangian simplifies greatly and we obtain the following definition of approximate KKT point for a minimization problem.

**Definition 3.** *Let $\varepsilon \geq 0$ be an approximation parameter, $f : \mathbb{R}^n \to \mathbb{R}$ a continuously differentiable function and consider the optimization problem $\min_{x \in [0,1]^n} f(x)$. The point $\widetilde{x} \in [0, 1]^n$ is an $\varepsilon$-approximate KKT point of the formulation if the gradient $g := \nabla f(\widetilde{x})$ satisfies for each $i \in [n]$:*

*1. If $\widetilde{x}_i \in (0, 1)$, then $|g_i| \leq \varepsilon$.*

*2. If $\widetilde{x}_i = 0$, then $g_i \geq -\varepsilon$.*

*3. If $\widetilde{x}_i = 1$, then $g_i \leq \varepsilon$.*

Thus, an exact ($\varepsilon = 0$) KKT point can be thought of as a fixed point of the gradient descent algorithm. Using this intuition, we can extend this to minmax problems as fixed points of the gradient *descent-ascent* algorithm. See Figure 2 for the geometric intuition of minmax KKT points.

**Definition 4.** *Let $\varepsilon \geq 0$ be an approximation parameter, $f : \mathbb{R}^n \times \mathbb{R}^n \to \mathbb{R}$ a continuously differentiable function and consider the optimization problem $\min_{x \in [0,1]^n} \max_{y \in [0,1]^n} f(x, y)$. Let $(\widetilde{x}, \widetilde{y}) \in [0, 1]^{2n}$ and let $(g, q) := \nabla f(\widetilde{x}, \widetilde{y})$, where $g$ is the gradient with respect to $x$-variables and $q$ with respect to $y$-variables. Then, $(\widetilde{x}, \widetilde{y})$ is an $\varepsilon$-approximate KKT point if for each $i \in [n]$:*

*1. If $x_i \in (0, 1)$, then $|g_i| \leq \varepsilon$.*   *4. If $y_i \in (0, 1)$, then $|q_i| \leq \varepsilon$.*

*2. If $x_i = 0$, then $g_i \geq -\varepsilon$.*   *5. If $y_i = 0$, then $q_i \leq \varepsilon$.*

*3. If $x_i = 1$, then $g_i \leq \varepsilon$.*   *6. If $y_i = 1$, then $q_i \geq -\varepsilon$.*

## 2.3 Connection between two-team games and minmax optimization

Given a two-team polymatrix zero-sum game with matrices $\{A^{i,j}\}_{i,j \in \mathcal{N}}$, we can define the following common utility function

$$U(x, y) = -\sum_{i,i' \in X : i < i'} x_i A^{i,i'} x_{i'} - \sum_{i \in X, j \in Y} x_i A^{i,j} y_j$$

and the corresponding game where players on the $X$-team all have the same utility function $-U$, and players on the $Y$-team all have the same utility function $U$. It is easy to check that this new game is

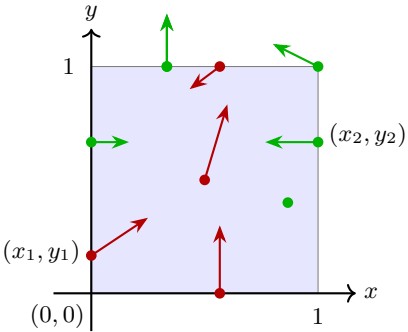

Figure 2: The intuition behind exact ($\varepsilon = 0$) KKT points for minmax problems. The formulation features a min-variable $x$, a max-variable $y$ and the bounding box constraint $(x, y) \in [0, 1]^2$. Some feasible points together with their respective gradient $(q^x, q^y)$ are depicted. Green points are valid KKT points whereas red ones are not. For instance, $(x_1, y_1)$ is not a KKT point because $q_1^y > 0$ but $y < 1$. On the other hand, $(x_2, y_2)$ is a valid KKT point because $q_2^y = 0$ and $q_2^x < 0$ with $x = 1$.

equivalent to the previous one, namely a strategy profile $(x, y)$ is an $\varepsilon$-Nash equilibrium in the latter game if and only if it is in the former. Now, if we consider the minmax optimization problem

$$\min_{x \in \mathcal{X}} \max_{y \in \mathcal{Y}} U(x, y)$$

it is not hard to show that its KKT points exactly correspond to the set of Nash equilibria of the game with common utility, and thus also of the original polymatrix game. This connection also extends to approximate solutions. We develop this relation in greater detail in section Section 3, where we first prove a hardness result for a minmax problem and then use the equivalence above to extend it to polymatrix games.

## 3  CLS-HARDNESS OF TWO-TEAM GAMES WITH INDEPENDENT ADVERSARIES

In this section, we give a proof of our main hardness result which we re-state formally next.

**Theorem 3.1** (Precise formulation of Theorem 1.1). *It is CLS-hard to find an $\varepsilon$-approximate Nash equilibrium of a two-team zero-sum polymatrix game with independent adversaries where $\varepsilon$ is inverse exponential in the description of the game.*

We show that this is true even when each player can only choose from two pure strategies. This lower bound result is obtained by reducing from the problem of finding some KKT point of a quadratic minimization problem. In detail, an instance of MinQuadKKT consists of a quadratic polynomial $Q : \mathbb{R}^n \to \mathbb{R}$ together with an approximation parameter $\varepsilon > 0$. A solution to such instance is any $\varepsilon$-approximate KKT point of $\min_{x \in [0,1]^n} Q(x)$ (see Definition 3). MinQuadKKT is known to be CLS-complete for $\varepsilon$ inverse exponential in the description of the instance (Fearnley et al., 2024). The reduction from MinQuadKKT to two-team games is performed in two stages which we describe next.

**Stage I.**  As a first step, we show how to reduce MinQuadKKT to the intermediate problem MinmaxIndKKT. An instance of MinmaxIndKKT consists of an approximation parameter $\varepsilon > 0$ together with a polynomial $M : \mathbb{R}^{2n} \to \mathbb{R}$ satisfying the following three properties:

1. $M$ is multilinear.
2. $M$ has degree at most 2.
3. $M(x, y)$ has no monomial of type $y_i y_j$ for $i, j \in [n]$.

In this section, we conveniently refer to those three conditions as the *independence* property. A solution to such an instance is any $\varepsilon$-approximate KKT point of $\min_{x \in [0,1]^n} \max_{y \in [0,1]^n} M(x, y)$ (see Definition 4). Let us highlight that this step already establishes the CLS-hardness of uncoupled minmax multilinear formulations for a simple class of objectives.

**Stage II.** We exploit the independence property of the polynomial generated by the first stage reduction to reduce it further to a two-team zero-sum polymatrix game with independent adversaries. This is achieved through generalizing the equivalence between zero-sum games for two players and some class of minmax formulations.

## 3.1 STAGE I: FROM QUADRATIC TO MULTILINEAR

**Lemma 3.1.** MinQuadKKT *reduces to* MinmaxIndKKT

By "reduces to", we mean the usual definition of polynomial-time TFNP reduction. We begin by describing the reduction in detail, then highlight that it yields an objective with the independence property and finally prove correctness. Let $Q : [0,1]^n \to \mathbb{R}$ and $\varepsilon > 0$ be the MinQuadKKT instance. Write the coefficients of $Q$ explicitly as follows:

$$Q(x) = q + \sum_{i \in [n]} q_i x_i + \sum_{i \neq j} q_{ij} x_i x_j + \sum_{i \in [n]} q_{ii} x_i^2$$

Let $Z \geq 1$ be an upper bound on the sum of the absolute values of all coefficients of $Q$ and note that since $Q$ is a quadratic polynomial, it holds that for every $x \in [0,1]^n$ and $i \in [n]$:

$$|Q(x)| \leq Z \quad \text{and} \quad \left| \frac{\partial Q(x)}{\partial x_i} \right| \leq 2Z \tag{1}$$

Fix $T := 10Z$ and $\eta := 2\varepsilon^2/Z$. For each min-variable $x_i$ of the MinQuadKKT instance, we introduce two min-variables $x_i$ and $x_i'$ and a max-variable $y_i$ in the MinmaxIndKKT instance. We also define the following multilinear "copy" gadget:

$$\text{COPY}(x_i, x_i', y_i) := \left( x_i' - x_i \cdot (1 - 2\eta) - \eta \right) \cdot (y_i - 1/2)$$

The role of COPY is to force $x_i$ to be close to $x_i'$ at any KKT point, thus effectively *duplicating* $x_i$ into $x_i'$. This allows us to remove square terms of the objective function. To make this formal, let $Q' : [0,1]^{2n} \to \mathbb{R}$ be a copy of $Q$ where every occurrence of the form $x_i^2$ is replaced by $x_i x_i'$:

$$Q'(x, x') := q + \sum_{i \in [n]} q_i x_i + \sum_{i \neq j} q_{ij} x_i x_j + \sum_{i \in [n]} q_{ii} x_i x_i'$$

The full formulation of the MinmaxIndKKT instance is stated below. We note that the objective $M$ of the formulation indeed satisfies the independence property.[3] The proof of Lemma 3.1 now follows from Claim 3.1.

$$\min_{x,x' \in [0,1]^n} \max_{y \in [0,1]^n} M(x, x', y) \quad \text{where} \quad M(x, x', y) := Q'(x, x') + \sum_{i \in [n]} T \cdot \text{COPY}(x_i, x_i', y_i)$$

**Claim 3.1.** *For any $\varepsilon \in (0, 1/13]$, if $(x, x', y)$ is an $(\varepsilon^2/Z)$-approximate KKT point of the MinmaxIndKKT instance, then $x$ is an $\varepsilon$-approximate KKT point of the MinQuadKKT instance.*

The proof relies crucially on $T$ being a large constant so that the copy gadgets dominate $Q'(x, x')$ and the objective $M$. This forces any KKT point $(x, x', y)$ to have $y \in (0,1)^n$ and ultimately $x \approx x'$. A second step shows that $\partial Q(x)/\partial x_i \approx \partial M(x, x', y)/\partial x_i$, which is enough to conclude that $x$ is a KKT point of the MinQuadKKT instance. Readers can find the full proof in Appendix A.1.

## 3.2 STAGE II: FROM MINMAX TO TWO-TEAM GAMES

To prove Theorem 1.1, we use stage I (Lemma 3.1) and show how to reduce an instance of MinmaxIndKKT to a two-team zero-sum game with independent adversaries. Let $\varepsilon > 0$ be the approximation parameter of the instance and its objective be $M : \mathbb{R}^{2n} \to \mathbb{R}$. Since $M$ has the independence property, we can explicitly write its coefficient as:

$$M(x, y) := \alpha + \sum_{i \in [n]} \beta_i x_i + \sum_{i \neq j} \gamma_{ij} x_i x_j + \sum_{i \in [n]} \zeta_i y_i + \sum_{i,j \in [n]} \theta_{ij} x_i y_j$$

---

[3]Additional dummy max-variables can be added to ensure that the number of min- and max-variables are the same.

We construct a polymatrix game with two teams, each consisting of $n$ players. In the first (cooperative) team, there is one player $a_i$ corresponding to each variable $x_i$. The second team consists of $n$ independent adversaries, where player $b_i$ corresponds to variable $y_i$. The intent is that an optimal strategy profile of the players roughly corresponds to a KKT point $(x, y)$ of $M$. As stated earlier, we reduce to a very restricted setting where each player only has two actions. We thus specify the utility matrices of the game as elements of $\mathbb{R}^{2 \times 2}$ with:

$$A^{a_i, a_j} = A^{a_j, a_i} = \begin{bmatrix} -\gamma_{ij} & 0 \\ 0 & 0 \end{bmatrix} \qquad \text{for all } i, j \in [n] \text{ with } i \neq j \quad (2)$$

$$A^{b_j, a_i} = -(A^{a_i, b_j})^\top = \begin{bmatrix} \theta_{ij} + \zeta_j/n + \beta_i/n & \zeta_j/n \\ \beta_i/n & 0 \end{bmatrix} \qquad \text{for all } i, j \in [n] \quad (3)$$

Any other utility matrix is set to $0^{2 \times 2}$. Observe that this payoff setting indeed yields a proper two-team zero-sum polymatrix game with independent adversaries. Let $Z \geq 1$ be an upper bound on the sum of absolute coefficients of $M$ and let $\delta := \varepsilon^2/(4Z)$ be the target approximation ratio for the polymatrix game. If $(p, q)$ is a $\delta$-approximate Nash equilibrium of the polymatrix game, we define a candidate KKT point $(x, y) \in \mathbb{R}^{2n}$ as follows:

$$x_i = \begin{cases} 0 & \text{if } p_i < \varepsilon/(2Z) \\ 1 & \text{if } p_i > 1 - \varepsilon/(2Z) \\ p_i & \text{else} \end{cases} \qquad \text{and} \qquad y_i = \begin{cases} 0 & \text{if } q_i < \varepsilon/(2Z) \\ 1 & \text{if } q_i > 1 - \varepsilon/(2Z) \\ q_i & \text{else} \end{cases}$$

Here $p_i \in [0, 1]$ represents the probability that player $a_i$ plays its first action, and similarly $q_i$ for player $b_i$. The correctness of the reduction is treated in Claim 3.2 and thus Theorem 1.1 follows.

**Claim 3.2.** $(x, y)$ *is an $\varepsilon$-approximate KKT point of the* MinmaxIndKKT *instance.*

*Proof.* We only show that $x$-variables satisfy the KKT conditions as the proof is similar for the $y$-variables. It follows from equation 2 and equation 3 that for any $\widetilde{p}_i \in [0, 1]$ the utility for player $a_i$ can be written as:

$$U_{a_i}(\widetilde{p}_i, p_{-i}, q) = \sum_{j \neq i} \begin{bmatrix} \widetilde{p}_i \\ 1 - \widetilde{p}_i \end{bmatrix}^\top A^{a_i, a_j} \begin{bmatrix} p_j \\ 1 - p_j \end{bmatrix} + \sum_{j \in [n]} \begin{bmatrix} \widetilde{p}_i \\ 1 - \widetilde{p}_i \end{bmatrix}^\top A^{a_i, b_j} \begin{bmatrix} q_j \\ 1 - q_j \end{bmatrix}$$

$$= -\widetilde{p}_i \cdot \left( \beta_i + \sum_{j \neq i} \gamma_{ij} p_j + \sum_{j \in [n]} \theta_{ij} q_j \right) - \sum_{j \in [n]} \zeta_j q_j / n$$

Since $(p, q)$ is a $\delta$-approximate Nash equilibrium, we can use the above expression twice and Definition 1 to get that:

$$(p_i - \widetilde{p}_i) \cdot \left( \beta_i + \sum_{j \neq i} \gamma_{ij} p_j + \sum_{j \in [n]} \theta_{ij} q_j \right) \leq \delta \quad \forall \widetilde{p}_i \in [0, 1] \quad (4)$$

Fix some variable $x_i$ and let us verify that it satisfies the $\varepsilon$-approximate KKT condition. The partial derivative of $M$ with respect to $x_i$ is:

$$\frac{\partial M(x, y)}{\partial x_i} = \beta_i + \sum_{j \neq i} \gamma_{ij} x_j + \sum_{j \in [n]} \theta_{ij} y_j$$

$$= \beta_i + \sum_{j \neq i} \gamma_{ij} \cdot (p_j + x_j - p_j) + \sum_{j \in [n]} \theta_{ij} \cdot (q_j + y_j - q_j)$$

$$= \beta_i + \sum_{j \neq i} \gamma_{ij} p_j + \sum_{j \in [n]} \theta_{ij} q_j \pm \left( \sum_{j \neq i} |\gamma_{ij}| \cdot |x_j - p_j| + \sum_{j \in [n]} |\theta_{ij}| \cdot |y_j - q_j| \right)$$

$$= \beta_i + \sum_{j \neq i} \gamma_{ij} p_j + \sum_{j \in [n]} \theta_{ij} q_j \pm \frac{\varepsilon}{2Z} \cdot Z$$

Where in the last equality, we used the fact that $x$ (respectively $y$) is close to $p$ (respectively $q$).

We now finish the proof by considering two cases for $x_i$. First, consider the case where $x_i < 1$. By definition of $x_i$, this implies that $p_i \leq 1 - \varepsilon/2Z$. Thus, setting $\widetilde{p}_i := 1$ in equation 4, we get:

$$(1 - p_i) \cdot \left( \beta_i + \sum_{j \neq i} \gamma_{ij} p_j + \sum_{j \in [n]} \theta_{ij} q_j \right) \geq -\delta$$

$$\implies \beta_i + \sum_{j \neq i} \gamma_{ij} p_j + \sum_{j \in [n]} \theta_{ij} q_j \geq -\frac{\delta}{1 - p_i} \geq -\frac{2Z \cdot \delta}{\varepsilon} \geq -\varepsilon/2$$

and thus $\partial M(x, y)/\partial x_i \geq -\varepsilon/2 - \varepsilon/2 \geq -\varepsilon$. Next, consider the case where $x_i > 0$. By definition of $x_i$, this implies that $p_i \geq \varepsilon/2Z$. Now, setting $\widetilde{p}_i := 0$ in equation 4, we get:

$$p_i \cdot \left( \beta_i + \sum_{j \neq i} \gamma_{ij} p_j + \sum_{j \in [n]} \theta_{ij} q_j \right) \leq \delta$$

$$\implies \beta_i + \sum_{j \neq i} \gamma_{ij} p_j + \sum_{j \in [n]} \theta_{ij} q_j \leq \frac{\delta}{p_i} \leq \frac{2Z \cdot \delta}{\varepsilon} \leq \varepsilon/2$$

and thus $\partial M(x, y)/\partial x_i \leq \varepsilon/2 + \varepsilon/2 \leq \varepsilon$. This shows that $x_i$ always satisfies the $\varepsilon$-KKT conditions. In particular, when $x_i \in (0, 1)$, $|\partial M(x, y)/\partial x_i| \leq \varepsilon$. $\qquad\square$

## 4 CLS-MEMBERSHIP FOR INDEPENDENT ADVERSARIES

In this section we prove the following.

**Theorem 4.1.** *The problem of computing a Nash equilibrium in two-team zero-sum polymatrix games with independent adversaries lies in* CLS.

In particular, this implies that the CLS-hardness result proved in the previous section is tight for such games.

**Reformulation as a minimization problem.** The main idea to prove the theorem is to start from the minmax formulation of the problem and to rewrite it as a minimization problem by using duality. Let a two-team polymatrix zero-sum game with independent adversaries be given. Without loss of generality, we assume that every player has exactly $m$ strategies. Recall that by the structure of the game we have $A^{i,i'} = (A^{i',i})^\top$, $A^{i,j} = -(A^{j,i})^\top$, and $A^{j,j'} = 0$ for all $i, i' \in X$, $j, j' \in Y$ with $i \neq i'$ and $j \neq j'$. We can write

$$
\begin{aligned}
& \min_{x \in \mathcal{X}} \max_{y \in \mathcal{Y}} - \sum_{i,i' \in X: i < i'} x_i A^{i,i'} x_{i'} - \sum_{i \in X, j \in Y} x_i A^{i,j} y_j \\
= & \min_{x \in \mathcal{X}} \max_{y \in \mathcal{Y}} - \sum_{i,i' \in X: i < i'} x_i A^{i,i'} x_{i'} + \sum_{i \in X, j \in Y} y_j A^{j,i} x_i \\
= & \min_{x \in \mathcal{X}} \left\{ - \sum_{i,i' \in X: i < i'} x_i A^{i,i'} x_{i'} + \max_{y \in \mathcal{Y}} \sum_{i \in X, j \in Y} y_j A^{j,i} x_i \right\}
\end{aligned}
\tag{5}
$$

Now consider the "max" part of the above objective written as the following LP in $y$ variables:

$$
\begin{aligned}
\max \quad & \sum_{j \in Y} c_j^\top y_j \\
& \sum_{k=1}^{m} y_{jk} = 1 \quad \forall j \in Y \\
& y_{jk} \geq 0 \quad \forall j \in Y \text{ and } \forall k \in [m]
\end{aligned}
$$

where $c_j = \sum_{i \in X} A^{j,i} x_i$ for all $j \in Y$. Then the dual of the above program can be written as:

$$
\begin{aligned}
\min \quad & \sum_{j \in Y} \gamma_j \\
& \gamma_j \geq c_{jk} \quad \forall j \in Y \text{ and } \forall k \in [m]
\end{aligned}
$$

Thus replacing the max part in equation 5 by the equivalent dual formulation we obtain:

$$
\begin{aligned}
\min \quad & -\sum_{i,i' \in X: i < i'} x_i A^{i,i'} x_{i'} + \sum_{j \in Y} \gamma_j \\
\text{s.t. } & \gamma_j \geq \sum_{i \in X} e_k^\top A^{j,i} x_i \ \ \forall j \in Y \ \text{ and } \ \forall k \in [m] \\
& \gamma_j \leq M \ \ \forall j \in Y \\
& \gamma \in \mathbb{R}^{|Y|} \\
& x \in \mathcal{X}
\end{aligned}
\tag{6}
$$

where $e_k \in \mathbb{R}^m$ is the $k$th unit vector. We have introduced an additional set of constraints $\gamma_j \leq M$ to ensure that the feasible region is bounded. $M$ is chosen to be sufficiently large such that $M > \max_{x \in \mathcal{X}, j \in Y, k \in [m]} \sum_{i \in X} e_k^\top A^{j,i} x_i$. Note that in order to obtain the formulation equation 6 we have crucially used the fact that the original game is polymatrix and has independent adversaries.

For what comes next, we will need the following definition of KKT points, which generalizes Definition 3 to arbitrary linear constraints.

**Definition 5.** *Consider an optimization problem of the form*

$$
\begin{aligned}
\min \quad & f(x) \\
\text{s.t. } & Ax \leq b \\
& x \in \mathbb{R}^n
\end{aligned}
$$

*where $f : \mathbb{R}^n \to \mathbb{R}$ is continuously differentiable, $A \in \mathbb{R}^{m \times n}$, and $b \in \mathbb{R}^m$. A point $x^* \in \mathbb{R}^n$ is a KKT point of this problem if there exists $\mu \in \mathbb{R}^m$ such that*

*1. $\nabla f(x^*) + A^\top \mu = 0$*

*2. $Ax^* \leq b$*

*3. $\mu \geq 0$*

*4. $\mu^\top (b - Ax) = 0$*

We can now continue with our proof of Theorem 4.1. The problem of finding an exact KKT point of equation 6 lies in CLS. Indeed, it is known that finding an approximate KKT point of such a program lies in CLS (Fearnley et al., 2022, Theorem 5.1), and, given that the objective function is a quadratic polynomial, an approximate KKT point (with sufficiently small approximation error) can be turned into an exact one in polynomial time (see, e.g., (Fearnley et al., 2024, Lemma A.1)).

Theorem 4.1 now simply follows from the following claim, the proof of which is in Appendix A.2.

**Claim 4.1.** *Given a KKT point of equation 6, we can compute a Nash equilibrium of the original game in polynomial time.*

## 5 OPEN PROBLEMS

- What is the complexity of the two-team polymatrix setting when there is a single adversary? Recently, Anagnostides et al. (2025) show CLS-hardness, with two players in the team and one adversary (players have non-constant number of actions). But, hardness of the two action case is still open. Note that CLS-membership still applies in these cases.

- Our CLS-membership result only applies to the setting where the adversaries are independent. If interactions between adversaries are allowed then the problem is only known to lie in PPAD, so there is a gap with the CLS-hardness that we show. Is the problem CLS-complete, PPAD-complete, or neither?

- Our hardness result provides strong evidence that no algorithm with running time $O(\text{poly}(\log(1/\varepsilon)))$ exists. However, there are gradient-based approaches that yield algorithms with running time $O(\text{poly}(1/\varepsilon))$ (can be adapted from the algorithm of Anagnostides et al. (2023)). What is the optimal polynomial dependence in $1/\varepsilon$ for such algorithms?

## ACKNOWLEDGMENTS

AH and GM were supported by the Swiss State Secretariat for Education, Research and Innovation (SERI) under contract number MB22.00026. SGN would like to thank Ioannis Panageas and colleagues at ZIB for discussions pertaining to this project.

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

# A MISSING PROOFS

## A.1 PROOF OF CLAIM 3.1

*Proof.* Let $\delta := \varepsilon^2/Z$ be the MinmaxIndKKT approximation guarantee for the remainder of this argument. As a first step, we show that $y_i \in (0, 1)$ for each $i \in [n]$. Towards contradiction, let us first suppose that $y_i = 0$ and observe that:

$$
\begin{aligned}
\frac{\partial M(x, x', y)}{\partial x_i} &= \frac{\partial Q'(x, x')}{\partial x_i} + T \cdot (1 - 2\eta) \cdot (1/2 - y_i) \\
&\geq \frac{\partial Q'(x, x')}{\partial x_i} + T \cdot (2/5) && \text{as } y_i = 0 \text{ and } \eta \leq 1/10 \\
&\geq -2Z + T \cdot (2/5) && \text{using equation 1} \\
&> \delta && \text{as } T = 10Z \text{ and } Z \geq 1
\end{aligned}
$$

But as $(x, x', y)$ is a $\delta$-approximate KKT point and $x_i$ is a min-variable, it must be that $x_i = 0$ (see Definition 4). A similar computation shows that if $y_i = 0$, then $x'_i = 1$ and so:

$$
\frac{\partial M(x, x', y)}{\partial y_i} = x'_i - x_i \cdot (1 - 2\eta) + \eta = 1 + \eta > \delta
$$

Because $y_i$ is a max-variable and $(x, x', y)$ a $\delta$-KKT point, it must be that $y_i = 1$: a contradiction to the assumption that $y_i = 0$. One can rule out the possibility of $y_i = 1$ in a similar way and we may thus conclude that $y_i \in (0, 1)$. This fact, together with the KKT conditions, further implies that the partial derivative of $M$ with respect to $y_i$ vanishes for each $i \in [n]$, and thus:

$$|x'_i - x_i \cdot (1 - 2\eta) - \eta| = \left| \frac{\partial M(x, x', y)}{\partial y_i} \right| \leq \delta \tag{7}$$

This shows that $x'_i \in [\eta - \delta, 1 - \eta + \delta]$ and combining with $\eta > \delta$ it follows that $x'_i \in (0, 1)$ and the KKT conditions imply:

$$\delta \geq \left| \frac{\partial M(x, x', y)}{\partial x'_i} \right| = \left| \frac{\partial Q'(x, x')}{\partial x'_i} + T \cdot (y_i - 1/2) \right| \implies T \cdot (1/2 - y_i) = \frac{\partial Q'(x, x')}{\partial x'_i} \pm \delta$$

Where we use the notation $a = b \pm \delta$ to mean $a \in [b - \delta, b + \delta]$. We can now show that the partial derivative of $M$ at $(x, x', y)$ is very close to the one of $Q$ at $x$ for all coordinates $x_i$:

$$\begin{aligned}
\frac{\partial M(x, x', y)}{\partial x_i} &= \frac{\partial Q'(x, x')}{\partial x_i} + (1 - 2\eta) \cdot T \cdot (1/2 - y_i) \\
&= \frac{\partial Q'(x, x')}{\partial x_i} + (1 - 2\eta) \cdot \left( \frac{\partial Q'(x, x')}{\partial x'_i} \pm \delta \right) \\
&= \frac{\partial Q'(x, x')}{\partial x_i} + \frac{\partial Q'(x, x')}{\partial x'_i} \pm \left( 2\eta \left| \frac{\partial Q'(x, x')}{\partial x'_i} \right| + \delta \right) \\
&= \frac{\partial Q'(x, x')}{\partial x_i} + \frac{\partial Q'(x, x')}{\partial x'_i} \pm (4\eta Z + \delta) \\
&= \frac{\partial Q'(x, x')}{\partial x_i} + \frac{\partial Q'(x, x')}{\partial x'_i} \pm 9\varepsilon^2 \qquad\qquad \text{as } \eta = 2\varepsilon^2/Z \text{ and } \delta \leq \varepsilon^2
\end{aligned}$$

Observe that equation 7 also implies that $x_i$ and $x'_i$ must be close with $|x_i - x'_i| \leq \delta + \eta$, hence:

$$\begin{aligned}
\frac{\partial Q'(x, x')}{\partial x_i} + \frac{\partial Q'(x, x')}{\partial x'_i} &= q_i + \sum_{j \neq i} q_{ij} x_j + q_{ii} x_i + q_{ii} x'_i \\
&= \frac{\partial Q(x)}{\partial x_i} + q_{ii}(x'_i - x_i) \\
&= \frac{\partial Q(x)}{\partial x_i} \pm 3\varepsilon^2 \qquad\qquad \text{as } |q_{ii}| \leq Z \text{ and } \delta + \eta = 3\varepsilon^2/Z
\end{aligned}$$

Combining the two previous observations, we have $\partial Q(x)/\partial x_i = \partial M(x, x', y)/\partial x_i \pm 12\varepsilon^2$. With this fact established, we may finally show that $x$ is indeed an $\varepsilon$-KKT point of the MinQuadKKT formulation. If $x_i = 0$, then note that:

$$\frac{\partial Q(x)}{\partial x_i} \geq \frac{\partial M(x, x', y)}{\partial x_i} - 12\varepsilon^2 \geq -\delta - 12\varepsilon^2 \geq -13\varepsilon^2 \geq -\varepsilon$$

A similar computation shows that the KKT conditions are also satisfied if $x_i = 1$. On the other hand, if $x_i \in (0, 1)$:

$$\left| \frac{\partial Q(x)}{\partial x_i} \right| \leq \left| \frac{\partial M(x, x', y)}{\partial x_i} \right| + 12\varepsilon^2 \leq \delta + 12\varepsilon^2 \leq \varepsilon. \qquad\qquad \square$$

## A.2 Proof of Claim 4.1

*Proof.* Let $(x^*, \gamma^*)$ be a KKT point of equation 6. We define notation for the following multipliers:

- For all $j \in Y$ and $k \in [m]$, $\mu^*_{jk} \in \mathbb{R}_{\geq 0}$ corresponding to the constraint $\gamma_j \geq \sum_{i \in X} e_k^\top A^{j,i} x_i$.

- For all $i \in X$, $\lambda^*_i \in \mathbb{R}$ corresponding to the constraint $\sum_{k \in [m]} x_{ik} = 1$.

- For all $i \in X$ and $k \in [m]$, $\nu^*_{ik} \in \mathbb{R}_{\geq 0}$ corresponding to the constraint $x_{ik} \geq 0$.

Note that we have not included multipliers for the constraints $\gamma_j \leq M$. This is because none of these constraints will ever be tight at a KKT point $(x^*, \gamma^*)$ by construction of $M$.

Now, since $(x^*, \gamma^*)$ is a KKT point, we can compute such multipliers that satisfy the following KKT conditions (which are derived from Definition 5) in polynomial time[4]:

1. For all $i \in X$

$$-\sum_{i' \neq i} A^{i,i'} x_{i'}^* - \sum_{j \in Y} A^{i,j} \mu_j^* + \lambda_i^* \cdot 1_m - \nu_i^* = 0$$

where $1_m$ denotes a vector of $m$ ones, and where we used the fact that $A^{i,j} = -(A^{j,i})^\top$. Additionally, we have that for all $k \in [m]$, $\nu_{ik}^* > 0 \implies x_{ik}^* = 0$.

2. For all $j \in Y$

$$1 - \sum_{k \in [m]} \mu_{jk}^* = 0.$$

3. For all $j \in Y$ and $k \in [m]$, $\mu_{jk}^* > 0 \implies \gamma_j^* = \sum_{i \in X} e_k A^{j,i} x_i^*$.

Now, we claim that $(x^*, \mu^*)$ forms a Nash equilibrium of the original game. First of all, note that by property 2 we have $\sum_{k \in [m]} \mu_{jk}^* = 1$ and thus $\mu_j^*$ is indeed a valid mixed strategy for player $j \in Y$. Next, property 3 can be rewritten as

$$\mu_{jk}^* > 0 \implies U_j(x^*, \mu_{-j}^*, e_k) = \sum_{i \in X} e_k A^{j,i} x_i^* = \gamma_j^* \geq \max_{k' \in [m]} \sum_{i \in X} e_{k'} A^{j,i} x_i^* = \max_{k' \in [m]} U_j(x^*, \mu_{-j}^*, e_{k'})$$

which means that the strategy of player $j$, namely $\mu_j^*$, is a best-response. Finally, property 1 can be reinterpreted as saying that $x_i^*$ is a KKT point of the following optimization problem (in variables $x_i$)

$$\min \quad -\sum_{i' \neq i} x_i A^{i,i'} x_{i'}^* - \sum_{j \in Y} x_i A^{i,j} \mu_j^* \quad [= U_i(x_i, x_{-i}^*, \mu^*)]$$
$$\text{s.t. } x_i \in \mathcal{X}_i$$

Since this is an LP, any KKT point is also a global solution. Thus, $x_i^*$ is a global minimum of this LP, which means that player $i$ is also playing a best-response. □

## B  ADDITIONAL RELATED WORK

**Linear Convergence for Bilinear/Convex-Concave Minmax**   Although we study this problem motivated by two-team polymatrix games with adversaries, the hardness results that we show also apply to the simplest non-convex concave minmax problem, i.e., bilinear non-convex concave. Our results strictly rule out the possibility of obtaining linear convergence. In contrast, bilinear zero-sum games and convex-concave games admit algorithms that converge linearly to the NE, for example see Wei et al. (2020); Lei et al. (2021); Sokota et al. (2022); Liu et al. (2022).

**Why two-team adversarial games/polymatrix games are interesting?**   The study of team games was initiated by von Stengel & Koller (1997), to model "imperfect" coordination within a company, when having to take strategic decisions in the presence of adversaries. In the field of AI agents, one can imagine such interactions are natural in settings where AI agents are trained to play team games, such as Starcraft (Vinyals et al., 2019) and DoTA (Berner et al., 2019).

Meanwhile, polymatrix games are used to model pairwise interactions between players and these interactions can be specified as a graph. In some cases, polymatrix games offer tractable alternative models for multiplayer games, as NE in polymatrix zero-sum games are efficiently computable (Daskalakis & Papadimitriou, 2009; Cai et al., 2016). More generally, polymatrix games are used to model problems such as coordination games on graphs (Apt et al., 2017; 2022) and this has applications in semi-supervised learning methods such as graph transduction (Erdem & Pelillo, 2011).

---

[4]Indeed, it is easy to check that given $(x^*, \gamma^*)$ such multipliers can then be found by solving an LP.

