# OpenReview forum: "The Complexity of Two-Team Polymatrix Games with Independent Adversaries"
_ICLR.cc/2025/Conference — ICLR 2025 Oral_

### Official Review · Reviewer_XrcJ · 2024-10-31

**Soundness:** 4
**Presentation:** 4
**Contribution:** 3
**Rating:** 8
**Confidence:** 3

**Summary:**

This paper studies the complexity of finding a Nash equilibrium in two-team polymatrix zero-sum games. They show that this problem is CLS-hard, and is in CLS if the adversaries are independent (thus establishing CLS-completeness in the latter case).

**Strengths:**

I think this is a good paper and vote to accept. The paper is clearly written and presents an interesting result. The hardness result about minimax KKT points is also rather clean, and may be of independent interest as a CLS-complete problem that may be relatively easy to make reductions from. The concerns below are very minor.

**Weaknesses:**

The section about ex-ante coordination contains some strange choices of phrasing. For example, all of the papers in that paragraph study extensive-form games (not just the last one), and the paper that shows "efficient algorithms exist under some assumptions about the players’ information set" is Zhang and Sandholm (2022), not Zhang et al. (2021).

To get parenthetical citations like (Lastname et al. 2023) instead of Lastname et al. (2023), use \citep.

**Questions:**

Perhaps the most obvious gap in this paper is the CLS-membership without the independent adversaries assumption. Do you think there is any hope to extend your techniques to that case?

---

> ### Author Response · Authors · 2024-11-22
>
> We thank the reviewer for their positive comments.
>
> Regarding the question about the possibility of extending the CLS-membership to the setting without the independent adversaries assumption: This is a very challenging question. It seems very unlikely that our technique for CLS-membership would extend to the setting without the independent adversaries assumption. On the other hand, the question of whether the min-max problem with uncoupled constraints is PPAD-hard is a major open question. Assuming that one could show PPAD-hardness for this general version of min-max, then we believe that our techniques could be used to extend this PPAD-hardness to two-team polymatrix games (without the independent adversaries assumption).
>
> We will fix the citation displays and now use \citep in appropriate locations.

---

> > ### Comment · Reviewer_XrcJ · 2024-11-26
> >
> > Thank you. My opinion of the paper has not changed, and I will keep my score.

---

### Official Review · Reviewer_xMtH · 2024-11-04

**Soundness:** 4
**Presentation:** 4
**Contribution:** 3
**Rating:** 8
**Confidence:** 3

**Summary:**

The paper studies the problem of finding Nash Equilibria in two-team polymatrix games. Polymatrix games are a special class of n-player games with a succinct representation of the payoff functions. Each player's payoff is a sum of payoffs resulting from two-player games played with all the other players. This problem is known to be tractable when all interactions are zero-sum, and to be PPAD-hard in general. A special subclass of these games are team games, where each pair of interactions are either zero-sum (different teams) or coordination games (same team). Three team games are known to be PPAD complete. The main result of the paper is in showing that two team games are CLS hard (CLS is a structured subclass of PPAD).  This result holds even when one of the team consists of independent adversaries (their games consist of the zero matrix for all the payoffs). They also show that computing the minimax/ KKT point of a bilinear polynomial is also CLS hard.

**Strengths:**

The paper solves a well formulated problem about the complexity of finding Nash equilibria. This problem is a natural continuation of prior results about team polymatrix games. The technical proofs and reductions are interesting and well written.

**Weaknesses:**

The main weakness is in the lack of appeal to a broader ICLR audience. The paper has solid results in complexity theory and game theory but requires some connection to the machine learning audience. That such a connection exists is not in itself in question, there are a plethora of papers about learning equilibria in team games, but the paper offers no discussion about the broader significance of studying team games. The open problems section also mentions gradient based methods that converge to equilibria in time poly(1/epsilon), but there is not further discussion.

**Questions:**

Could you add some discussion about the broader landscape of team games, why we might care about them (if not necessarily the two-player polymatrix team games), and about the best-known algorithmic results in this space, particularly in the context of learning dynamics?

---

> ### Author Response · Authors · 2024-11-22
> **Response Part I**
>
> We thank the reviewer for their positive comments.
>
> Regarding the relevance to a broader ICLR audience, our work has a multi-fold relevance to the ML audience who are broadly interested in minmax optimization and algorithmic game theory.
>
> ### Relevance to minmax optimization
> Although we study this problem motivated by two-team polymatrix games with adversaries, the hardness results that we show also apply to the _simplest_ non-convex concave minmax problem, i.e., bilinear non-convex concave. Our results strictly rule out the possibility of obtaining linear convergence. In contrast, bilinear zero-sum games and convex-concave games admit algorithms that converge linearly to the NE, for example see [Wei et al., 2020, Sokota et al., 2022, Mingyang et al., 2022] (papers that appeared at ICLR).
>
> Since the advent of GANs [Goodfellow et al., 2020], there has been a natural interest to study the best algorithms for it and as a result clearly understand the complexity of non-convex non-concave minmax [Daskalakis et al., 2021, Daskalakis 2021]. The current known PPAD-hardness results for this setting require _coupled_ constraints, which is unnatural for games or GANs. Our hardness results, on the other hand, are applicable to natural settings where the constraints are uncoupled, making it a ``natural'' nonconvex-concave minmax problem with known hardness.
> Moreover, our completeness results for the polymatrix setting poses an interesting question about the algorithms that achieve the best dependence on $(1/\varepsilon)$. We show that our minmax problem can be reduced to a minimization problem and note that to compute a NE, we require convergence only to a _first-order stationary point_.
> This minimization problem has connections to numerous practical problems such as Quadratic Assignment problems, Power flow optimization, Portfolio Optimization etc.

---

> > ### Comment · Reviewer_xMtH · 2024-11-25
> >
> > Thanks for the considered response. It has helped me situate the significance of the hardness result, specifically about this being the easiest non-convex concave minimax problem that has a hardness result blocking linear convergence.

---

> ### Author Response · Authors · 2024-11-22
> **Response Part II**
>
> ### Why two-team adversarial games/polymatrix games are interesting?
>
> The study of team games was initiated by [von Stengel and Koller 1997], to model ``imperfect'' coordination within a company, when having to take strategic decisions in the presence of adversaries. In the field of AI agents, one can imagine such interactions are natural in settings where AI agents are trained to play team games, such as Starcraft [Vinyals et al., 2019] and DoTA [Berner et al., 2019].
>
> Meanwhile, polymatrix games are used to model pairwise interactions between players and these interactions can be specified as a graph. In some cases, polymatrix games offer tractable alternative models for multiplayer games, as NE in polymatrix zero-sum games are efficiently computable [Daskalakis and Papadimitriou 2009, Cai et al., 2016]. More generally, polymatrix games are used to model problems such as coordination games on graphs [Apt et al., 2017, Apt et al., 2022] and this has applications in semi-supervised learning methods such as graph transduction [Aykut and Pelillo 2012, Vascon et al., 2020].
>
> Finally, as we state, our lower bounds automatically apply to the multiagent reinforcement learning setting, since our games are stateless.
>
> ### On Gradient Based Algorithms
>
> Our setting of two-team polymatrix games can be viewed as a special case of the two-team adversarial games studied by [Anagnostides et al., 2023]. Although, they describe their GradientDescentMAX algorithm for a single adversary, it can be applied to the case of many independent adversaries. Using their algorithm on the minmax objective gives us the $O(poly(size).1/\varepsilon^4)$ convergence rate to an $\varepsilon$-approximate NE.
>
>
>
>
> References:
> Goodfellow, Ian, et al. "Generative adversarial networks." Communications of the ACM 63.11 (2020): 139-144.
>
> Daskalakis, Constantinos. "Non-concave games: A challenge for game theory’s next 100 years." Nobel symposium” One Hundred Years of Game Theory: Future Applications and Challenges. 2021.
>
> Daskalakis, Constantinos, Stratis Skoulakis, and Manolis Zampetakis. "The complexity of constrained min-max optimization." Proceedings of the 53rd Annual ACM SIGACT Symposium on Theory of Computing. 2021.
>
> Cai, Yang, et al. "Zero-sum polymatrix games: A generalization of minmax." Mathematics of Operations Research 41.2 (2016): 648-655.
>
> Daskalakis, Constantinos, and Christos H. Papadimitriou. "On a network generalization of the minmax theorem." International Colloquium on Automata, Languages, and Programming. Berlin, Heidelberg: Springer Berlin Heidelberg, 2009.
>
> Erdem, Aykut, and Marcello Pelillo. "Graph transduction as a noncooperative game." Neural Computation 24.3 (2012): 700-723.
>
> Sokota, Samuel, et al. "A unified approach to reinforcement learning, quantal response equilibria, and two-player zero-sum games." arXiv preprint arXiv:2206.05825 (2022).
>
> Wei, Chen-Yu, et al. "Linear last-iterate convergence in constrained saddle-point optimization." arXiv preprint arXiv:2006.09517 (2020).
>
> Liu, Mingyang, et al. "The power of regularization in solving extensive-form games." arXiv preprint arXiv:2206.09495 (2022).
>
> Apt, Krzysztof R., Sunil Simon, and Dominik Wojtczak. "Coordination games on weighted directed graphs." Mathematics of Operations Research 47.2 (2022): 995-1025.
>
> Apt, Krzysztof R., et al. "Coordination games on graphs." International Journal of Game Theory 46 (2017): 851-877.
>
> Vinyals, Oriol, et al. "Grandmaster level in StarCraft II using multi-agent reinforcement learning." nature 575.7782 (2019): 350-354.
>
> Berner, Christopher, et al. "Dota 2 with large scale deep reinforcement learning." arXiv preprint arXiv:1912.06680 (2019).
>
> Anagnostides, Ioannis, et al. "Algorithms and complexity for computing nash equilibria in adversarial team games." arXiv preprint arXiv:2301.02129 (2023).

---

### Official Review · Reviewer_m3Yt · 2024-11-05

**Soundness:** 4
**Presentation:** 3
**Contribution:** 3
**Rating:** 8
**Confidence:** 4

**Summary:**

In this work, the authors investigate the computational complexity of computing a Nash equilibrium in two-team zero-sum polymatrix games where one team consists of independent players (i.e., players who do not interact with one another). Specifically, they prove that this problem is complete for the complexity class CLS. To demonstrate hardness, they first reduce from MinQuadraticKKT—the problem of computing a KKT point of a quadratic function with box constraints—to MinmaxIndKKT, a min-max problem with an independence property they define. In a second step, they reduce this problem to a two-team zero-sum polymatrix game. Membership in CLS follows fairly straightforwardly from the recent result that QuadraticKKT is complete for the class CLS, as shown in [1], and using LP duality for transforming a min-max problem into a minimization.

**Strengths:**

The paper is generally well-written, though there are areas where phrasing could be improved. The problem under consideration is quite interesting and represents a step forward in establishing complexity results for two-team (zero-sum) games, i.e., min-max optimization problems beyond the case of having coupled constraints as in [2]. Of course, the more general case of having dependent adversaries (or that of a single adversary) remains open, as the authors highlight in Section 5.

**Weaknesses:**

I cannot identify any obvious weaknesses. Although the techniques and ideas are not particularly complex—as is often the case in results of this kind—this should not in itself be considered a weakness. However, the simplicity of the proof and the lack of novel ideas makes me more skeptical about my final score.

**Questions:**

- **Line 386**: In the reduction from MinmaxIndKKT, the authors define the candidate KKT point $(x_i, y_i)$ for the case where neither $x_i$ nor $y_i$ is in ${0, 1\}$ as $x_i = a_i$ and $y_i = d_i$. I assume that $a_i$ is simply a typo, as $a_i$ is already used to denote player $i$ on the first team. I think the authors likely intended to use $p_i$ and $q_i$ for $x_i$ and $y_i$, which would also align with the statement in line 415 indicating that these variables are close to their respective counterparts, $p_i$ and $q_i$.

References
[1] The complexity of computing KKT solutions of quadratic programs.

[2] Constantinos Daskalakis, Stratis Skoulakis, and Manolis Zampetakis. The complexity of constrained min-max optimization.

---

> ### Author Response · Authors · 2024-11-22
>
> We thank the reviewer for their positive comments.
>
> Regarding the concern about the simplicity of the proof and the lack of novel ideas: Although the proof turns out to be quite simple, coming up with the construction in Lemma 3.1 is not trivial. This part required a novel construction that takes advantage of the max-variables to impose an additional constraint on the min-variables, which is crucial to get a bilinear objective and make the reduction work. This is the main novel technical contribution of our paper.
>
> Thank you for telling us about the typo on line 386. We will fix it in the updated version.

---

### Author Response · Authors · 2024-11-28
**Revised Draft**

We once again thank the reviewers for their positive comments. We have uploaded a revised draft including these clarifications and fixed typos (the changes are indicated in red).

---

### Meta-Review · Area_Chair_jr6Y · 2024-12-20

**Metareview:**

This paper looks at the complexity of computing teams equilibria in game.

It is a good theoretic contribution that the reviewers and I enjoyed reading. It is well written and rather insightful.

Happy to recommend acceptance.

**Additional Comments On Reviewer Discussion:**

All positive reviews and score, no discussion

---

### Decision · Program_Chairs · 2025-01-22

Accept (Oral)